# Characterization and field application of a novel dual-liquid gas leakage material: Mechanical properties and microscopic hydration mechanism

**Zijie Hong**[1,2], **Jianping Zuo**[3,4]*, **Zhenhua Li**[1], **Lei Xu**[1]

**1** School of Civil Engineering, Henan Polytechnic University, Jiaozuo, China, **2** The Project of Henan Key Laboratory of Underground Engineering and Disaster Prevention (Henan Polytechnic University), Jiaozuo, China, **3** School of Mechanics and Civil Engineering, China University of Mining and Technology (Beijing), Beijing, China, **4** School of Energy Science and Engineering, Henan Polytechnic University, Jiaozuo, China

☯ These authors contributed equally to this work.
¤ Current address: Henan Polytechnic University, Jiaozuo, China
‡ LX also contributed equally to this work.
* zjp@cumtb.edu.cn

**Data Availability Statement:** All relevant data are within the paper and its Supporting Information files.

## Abstract

Gas drainage materials are one critical aspect of preventing coal mine gas explosions. Here, a novel dual-liquid gas sealing material was developed to improve gas extraction. The mechanical properties and hydration mechanism of the proposed material were determined. The novel dual-liquid gas sealing material's performance was verified experimentally and with field testing, with practical application explored in the YunGaiShan 2 coal mine. The results showed that the main factor responsible for gas drainage leakage was the poor sealing effect of the sealing materials on the cracks around the borehole. The novel dual-liquid gas sealing material reduced damage to the rock surrounding the borehole and significantly improved the gas drainage performance. The initial and final setting times of the novel dual-liquid material were shown to be controllable, and the slurry exhibited good fluidity, with a 28-day uniaxial compressive strength of 11.06 MPa. The analysis of the microscopic hydration mechanism showed that the production of ettringite (AFt) in the dual-liquid material increased significantly, forming a denser network interlace that functioned as a network skeleton, improving the compressive strength of the material and achieving the characteristics of plastic deformation. Field-based analysis was performed to verify the practical applicability of the proposed material, showing that the gas drainage concentration increased by 200.5% compared to the original sealing material. Moreover, the average gas drainage negative pressure increased from 7.8 kPa (using the conventional sealing technique) to 16.6 kPa using the proposed material. Overall, the proposed materials are suitable for sealing materials for effective gas drainage performance and can help control gas disasters.

**Funding:** This study was supported by the Science and Technology Project of Henan Province (222102320004), the project of Henan Key Laboratory of Underground Engineering and Disaster Prevention (Henan Polytechnic University) (722403/018/001), the National Natural Science Foundation of China (52174073) and the Outstanding Young Scientist of Beijing (BJJWZYJH01201911413037).

**Competing interests:** NO authors have competing interests.

## Introduction

Coal consumption is essential in the global energy infrastructure and accounted for 27% of global energy consumption in 2020 [1]. The depth of coal mining in China is increasing by an average of 10 m/year [2]. The increasing depth results in higher gas pressures, higher contents of coal seams, and higher coalbed methane discharge during exploitation. Gas accidents at mines are one of the most severe threats to human and capital resources [3], affecting both the production of coal and personnel safety [4]. Coal seam gas is the main reason for gas outbursts and is released due to mining disturbance [5, 6]. Gas is a by-product of coal formation, and most of the gas is absorbed by coal seam [7]. Furthermore, coal seam gases are a source of methane released into the atmosphere and have 20 times greater effect than carbon dioxide as a greenhouse gas [8]. However, due to the characteristic presence of large fractures around boreholes and the low permeability and high absorptivity of coal seams, it is often difficult to extract gas from mines in China [9]. Gas accidents cause substantial property losses, endanger the lives of its people, and diminish mine safety and efficient production of resources.

Gas drainage is the most widely used treatment method and is considered a safe and efficient approach [10]. With the continuous improvement of gas extraction technology, the amount of extracted gas and its utilization are also increasing significantly. However, the amount of gas utilization is not proportional to the amount of gas extracted, and its growth rate is low. The sealing quality of the borehole affects the drainage of gas, with gas sealing materials reducing the borehole leakage and strengthening the boreholes to reduce deformation and damage [11, 12], which ensures adequate gas-drainage performance. Therefore, selecting a suitable gas sealing material is essential to ensure effective gas-drainage performance.

In China, the pre-drainage of coal seam gas by drilling is an effective method to prevent coal mine gas disasters and allows for the effective utilization of associated coal seam resources [13, 14]. Sealing coal seam boreholes has been one of the biggest problems encountered in gas drainage. Various scholars have studied gas drainage methods. In particular, Wang et al. studied the gas drainage technique under high gas and low permeability conditions [15], while Chen et al. experimentally determined the variation patterns of damage and permeability in coal [16]. Kong et al. [17] studied the problem of gas drainage in low permeability reservoir environments, showing that the gas drainage effect can be improved by adopting large-range gas drainage borehole technologies. Taheri studied the influence of microscopic changes in coal samples on gas damage [18], while Durucan established the working face permeability and gas flow model, outlining the variation patterns of coal stress and permeability [19]. Alam et al. [20] studied the influence of confining pressure on coal seam permeability, which is highly beneficial to the improvement of gas drainage. Wang et al. [21] analyzed the mechanism of coal seam leakage and proposed a gas extraction method using an expansive sealing material. Xiang et al. reported a flexible gel sealing material, which they used to propose a novel active sealing method [22]. Wang et al. studied the air leakage problem of cracks around gas extraction drilling holes by introducing the air leakage ring [23] and put forth a theoretical analysis to determine the size of the air leakage ring. Danesh et al. improved the permeability model of a coal seam, introduced the Nishihara model according to the anisotropic porous elastic media of surrounding rock, obtained the key factors affecting the gas drainage, and evaluated the gas drainage performance of coal seam [24].

Theoretically, when coal seam gas is extracted using borehole drilling, 100% of the gas should be extractable through drill holes with effective sealing. However, in practice, gas leakage results in less than 30% of gas drained from the boreholes, which seriously impacts the gas drainage performance. Many scholars have studied novel sealing materials to improve the

effect of gas extraction. Wang et al. [25] studied the double-liquid pore-sealing cement material and obtained better extraction concentration in gas extraction. Wang et al. [26] studied the sealing material of a cement bag and explained the sealing mechanism of gas extraction drilling. Xiang et al. [27] studied flexible gel gas extraction materials and reported that the flexible gel material had adaptability capabilities, and its properties, such as compactness and fluidity, were relatively good. Zhang et al. [28] introduced the application of polyurethane material in gas extraction, mainly using the polyurethane foaming mechanism to fill the cracks around the borehole, and reported that the material had the advantages of high foaming rate, easy penetration into the cracks in the hole wall, and no crack buildup.

However, the commercial application of current gas drainage methods is hindered by the poor performance of gas drilling sealing materials and low gas drainage concentrations [29, 30], which eventually leads to a rapid decrease in the gas drainage flow rate [31–33].

Generally, it is believed that the higher the strength of the gas extraction material, the better. However, the mechanism of coal seam gas leakage has not been systematically studied in the literature. The polymer materials commonly used in coal mines have complicated operation problems, are corrosive, and have large consumption and high cost. Inorganic cement materials for cement systems may have low costs and are widely available; however, common cement has a long setting time and undergoes cracking after solidification. Therefore, developing sealing materials that adequately seal gas leakage channels is critical for efficient and effective production.

Effective sealing materials should have the capacity to spread along cracks surrounding the holes and form a better seal to ensure no gap is left between the borehole wall and the sealing material. In addition, to avoid the development of secondary air leakage channels, the sealing material should have the ability to expand, supporting the drill hole and improving its ability to resist deformation. Because of the development of gas leakage channels around gas boreholes and the need to achieve better gas drainage effects, a novel dual-liquid gas sealing material was developed. The proposed sealing material's mechanical properties and hydration mechanism were studied. Finally, the performance of the novel dual-liquid gas sealing material was verified using field application.

## Experimental

### Material composition

The sealing material was composed of sulphoaluminate cement clinker, sulfur-building gypsum, and quicklime. The auxiliary additives consisting of a suspension agent, retarder, thickener and compound accelerator were also added to the sealing material. Sulphoaluminate cement clinker has the advantages of high early strength, high overall strength, good erosion resistance, impermeability, and frost resistance. Sulfur-building gypsum and quicklime improve the compressive strength of cement materials. The retarder is adsorbed on the cement surface and slows the cement hydration process, thus extending the slurry loss time. A thickener increases the viscosity of the slurry, improves the anti-segregation performance, and enhances the strength of the consolidated body. The materials used in the current work are shown in Figs 1 to 4.

The orthogonal test was designed to include four variable factors and three levels. The four factors included: A) the compound quick-setting agent, B) the compound retarder, C) the expansive agent, D) the water-cement ratio. Although the relative amount of auxiliary additives added to the mixture is minor, these compounds play an essential role in optimizing and improving performance. Therefore, these experiments focused mainly on the influence of

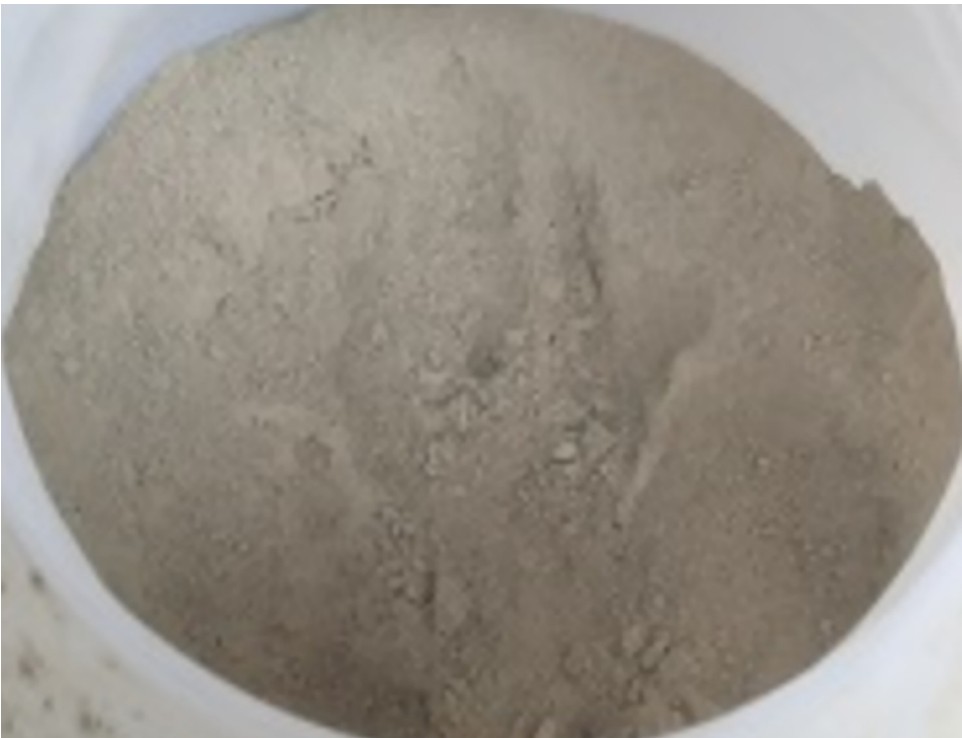

**Fig 1. Sulphoaluminate cement clinker.**

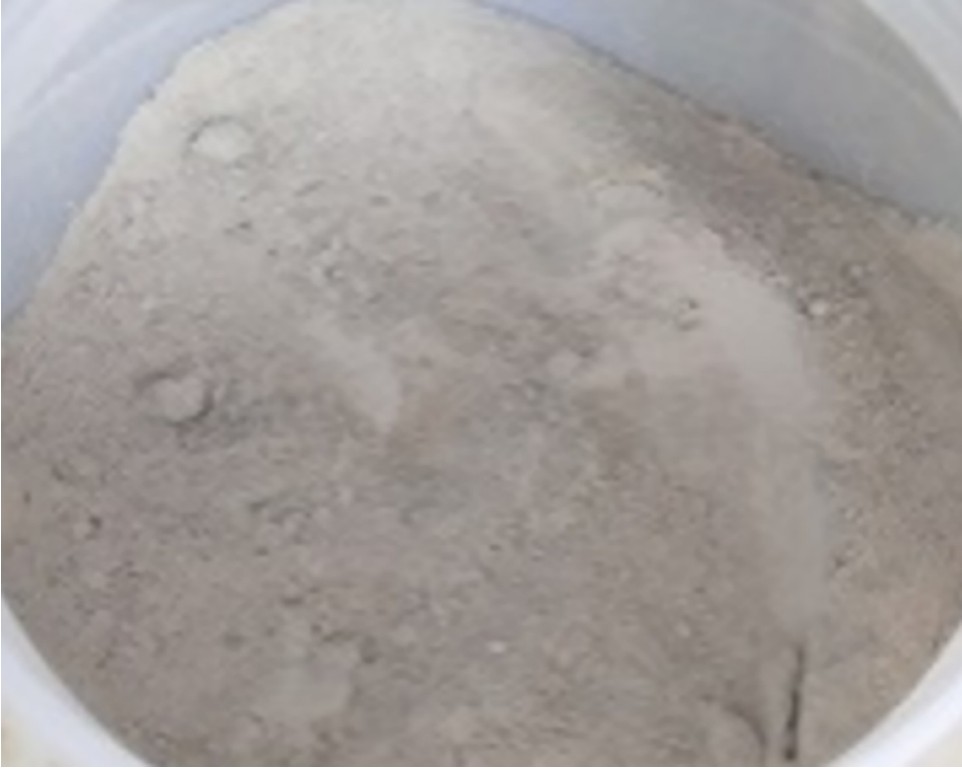

**Fig 2. Sulfur building gypsum and quicklime.**

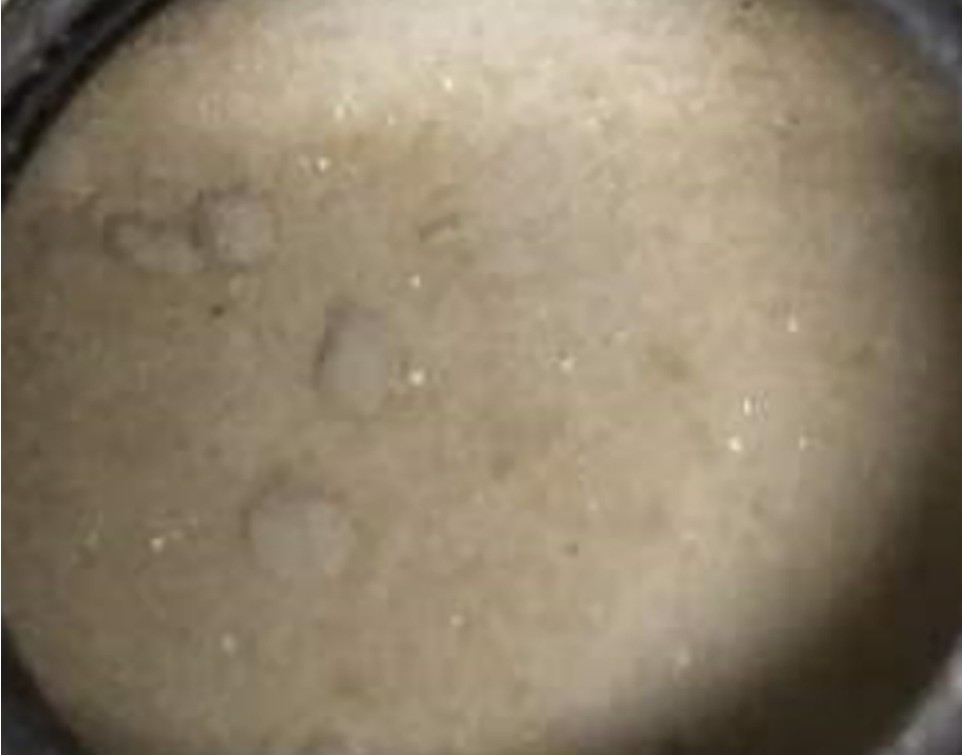

**Fig 3. Retarder.**

auxiliary additive components on the performance of the proposed material. Moreover, based on the results, the optimum contents of these additives are also determined.

## Parameters to gauge the material's performance

The measured parameters included the time required for gelation, the time required for final setting, fluidity and compressive strength, expansion performance, the microstructure, and the stress-strain mechanical properties of the material. The time required for gelation and the final setting was determined using the Vicat test, while the fluidity was measured using a slurry flow rate test. The microstructure of the sealing material was observed using scanning electron microscopy (SEM). Furthermore, conventional uniaxial, triaxial compression tests and damage fracture rock specimen preparation tests were performed using an RMT-150B rock mechanics testing system.

## Determination of gypsum and lime ratio

The properties of the sealing material have a significant influence on its compressive strength. Therefore, to determine the reasonable proportion of the main material (Y), the main ingredient (X) and compound agent were initially kept constant. Then, the proportion and total quantity of Y and the compound agent were fixed, while the ratio of gypsum to lime was varied. Meanwhile, the water-cement ratio was kept constant at 1.2:1.0. The corresponding results are presented in Table 1 and Fig 5.

As shown in Fig 2, for the gypsum lime ratios of less than 4:1, the uniaxial compressive strength increased. However, the uniaxial compressive strength decreased with the increased

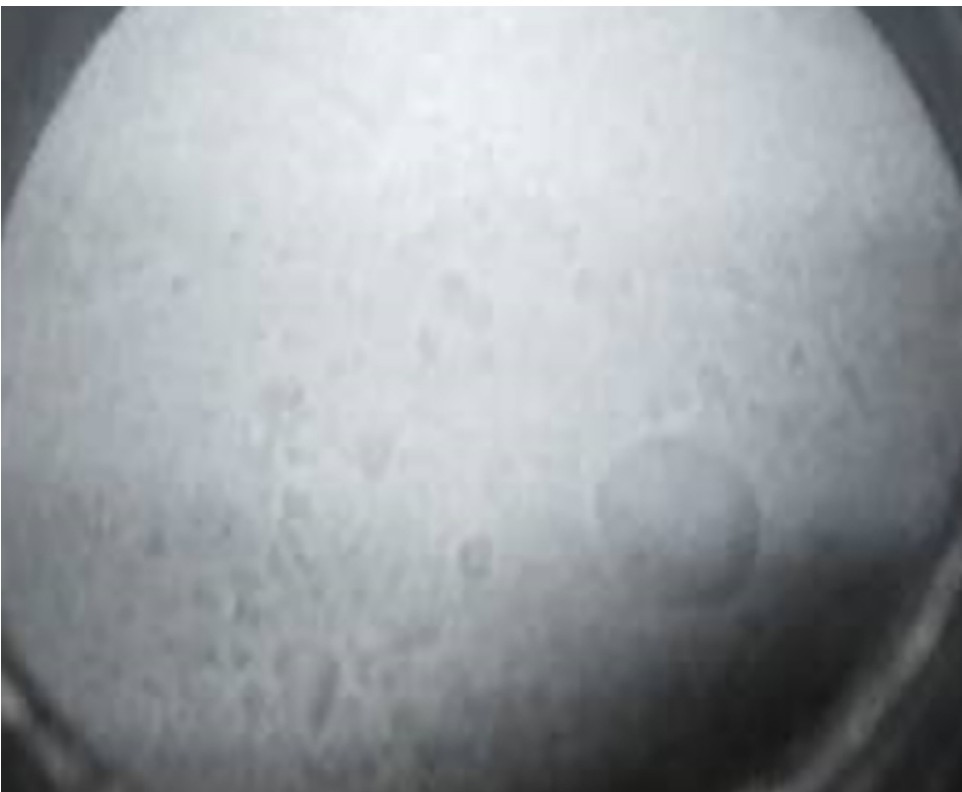

**Fig 4. Thickener.**

proportion of gypsum to lime in Groups 1–4. A gypsum-to-lime ratio of 4:1 achieved the maximum compressive strength, indicating that the primary material significantly influenced the uniaxial compressive strength. Furthermore, the other performance indicators can be adjusted by altering the components added to the sealing material. The optimal proportion of gypsum to lime in the sealing material was selected as 4:1.

## Results and discussion

Based upon a four-factor, three-level system, the effect of various factors, including compound quick-setting agent (A), compound retarder (B), expansive agent (C), and water-cement ratio (D) were investigated using an orthogonal experimental design. The orthogonal experimental design is presented in Table 2.

In this study, the mean value of each factor at each level was calculated according to the orthogonal system. The experimental results were based on the evaluation of gelling time, final

**Table 1. Different proportions of gypsum and lime and the effect of these variations on the compressive strength of the sealing material.**

| No. | Percentage of mass /% | | | | Compressive strength /MPa | | |
|---|---|---|---|---|---|---|---|
| | Gypsum | Lime | Complex agent | Expansion agent | 2h | 1d | 3d |
| 1–1 | 70 | 30 | 2 | 0.02 | 3.59 | 5.66 | 6.15 |
| 1–2 | 75 | 25 | 2 | 0.02 | 3.01 | 3.69 | 5.37 |
| 1–3 | 80 | 20 | 2 | 0.02 | 4.52 | 6.28 | 7.59 |
| 1–4 | 85 | 15 | 2 | 0.02 | 4.69 | 5.31 | 5.28 |

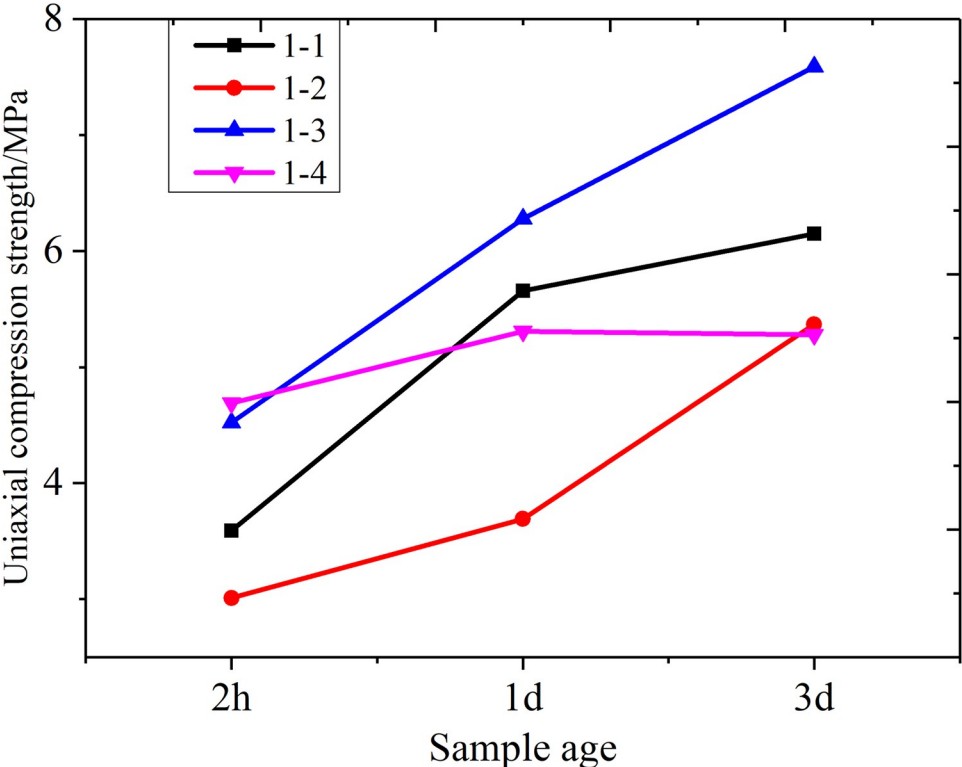

**Fig 5. Compressive s compressive strength curves of the sealing material at different curing ages.**

setting time, flow rate, expansion rate, and compressive strength. Table 3 presents the results of the orthogonal analysis of the proposed material.

Based upon the influence of optimal sealing material ratio on the performance of the proposed sealing material, the optimum proportions were determined to be 0.3% compound quick-setting agent, 0.5% compound retarder, 0.03% expansive agent and a water-cement ratio of 1.2:1.0. After determining the optimal component ratio, the initial and final setting times of the sealing material were determined to be 9 min and 23 min, respectively, resulting in a density of 1.25 g/cm$^3$, an expansion rate of 6.8%, fluidity of 336 mm and a 28-d compressive strength of 11.06 MPa.

**Table 2. Orthogonal (3$^4$) experimental design.**

| No. | Influencing factors | | | |
|---|---|---|---|---|
| | **A (%)** | **B (%)** | **C (%)** | **D** |
| $S_1$ | $A_1(0.1)$ | $B_1(0.3)$ | $C_1(0.02)$ | $D_1(1.0)$ |
| $S_2$ | $A_2(0.1)$ | $B_2(0.5)$ | $C_2(0.03)$ | $D_2(1.2)$ |
| $S_3$ | $A_3(0.1)$ | $B_3(0.7)$ | $C_3(0.05)$ | $D_3(1.5)$ |
| $S_4$ | $A_4(0.3)$ | $B_4(0.3)$ | $C_4(0.03)$ | $D_4(1.5)$ |
| $S_5$ | $A_5(0.3)$ | $B_5(0.5)$ | $C_5(0.05)$ | $D_5(1.0)$ |
| $S_6$ | $A_6(0.3)$ | $B_6(0.7)$ | $C_6(0.02)$ | $D_6(1.2)$ |
| $S_7$ | $A_7(0.5)$ | $B_7(0.3)$ | $C_7(0.05)$ | $D_7(1.2)$ |
| $S_8$ | $A_8(0.5)$ | $B_8(0.5)$ | $C_8(0.02)$ | $D_8(1.5)$ |
| $S_9$ | $A_9(0.5)$ | $B_9(0.7)$ | $C_9(0.03)$ | $D_9(1.0)$ |

**Table 3. Results of orthogonal analysis.**

| No. | Initial setting time /min | Final setting time /min | Fluidity/mm | Expansion ratio /% | Compressive strength /MPa | | |
|-----|---------------------------|-------------------------|-------------|---------------------|---------------------------|---|---|
| | | | | | 2h | 3d | 28d |
| S$_1$ | 7 | 15 | 290 | 4.3 | 5.61 | 8.92 | 15.85 |
| S$_2$ | 9 | 26 | 355 | 7.2 | 4.19 | 5.27 | 10.51 |
| S$_3$ | 17 | 40 | 385 | 11.4 | 2.31 | 4.02 | 7.09 |
| S$_4$ | 14 | 25 | 330 | 10.2 | 3.26 | 4.64 | 9.36 |
| S$_5$ | 5 | 11 | 305 | 12.3 | 3.00 | 7.28 | 13.88 |
| S$_6$ | 6 | 17 | 312 | 5.7 | 5.12 | 9.07 | 14.30 |
| S$_7$ | 8 | 22 | 326 | 13.6 | 5.59 | 5.57 | 8.72 |
| S$_8$ | 10 | 19 | 346 | 5.4 | 3.68 | 5.89 | 9.79 |
| S$_9$ | 4 | 10 | 285 | 7.2 | 6.83 | 10.22 | 18.43 |

## Micromorphology analysis

Scanning electron microscopy (SEM) was used to analyze the micro-morphology of the hydration product in detail. The SEM images of the hydrated material are shown in Figs 6 to 11. It was observed that the main hydration product was ettringite (AFt). A uniform surface was observed, with a series of grooves and openings (Figs 6 and 7). Samples collected after 2h (Fig 6) and 28d (Fig 7) exhibited more air bubbles on their surfaces, especially after 28d. Both the surface bubbles and pore size increased significantly with the progression of time. As the expansion agent produced bubbles and calcium aluminate hydrate in alkaline environments as a continuous hydration process, ongoing aging of the material allowed for increased hydration, resulting in the development of more bubbles. In the gypsum environment, calcium

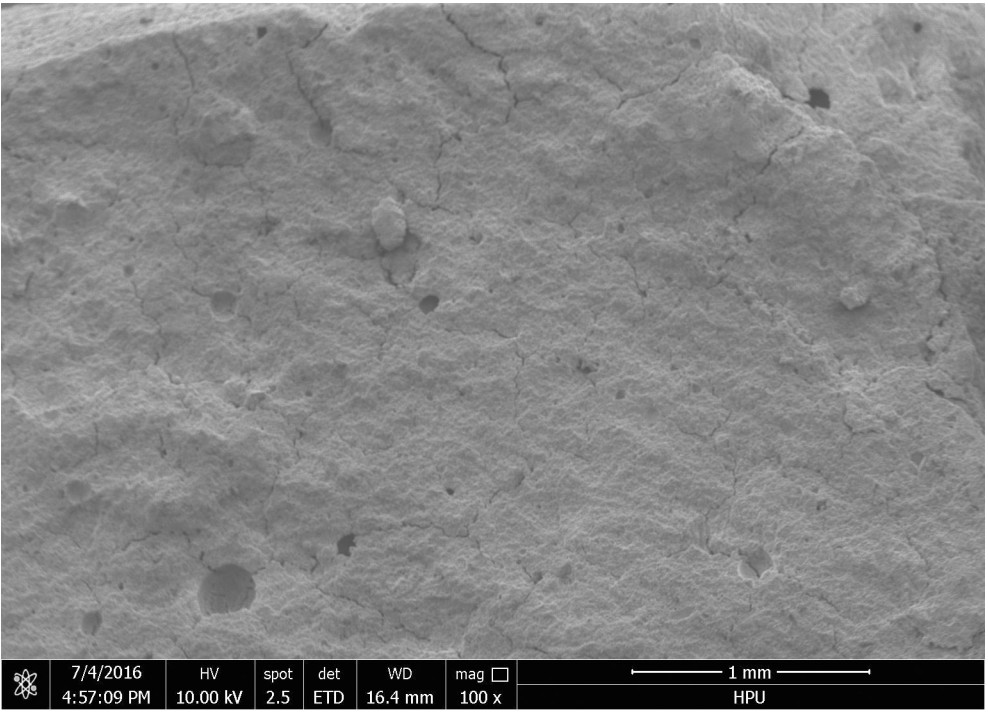

**Fig 6. 3d (×100).**

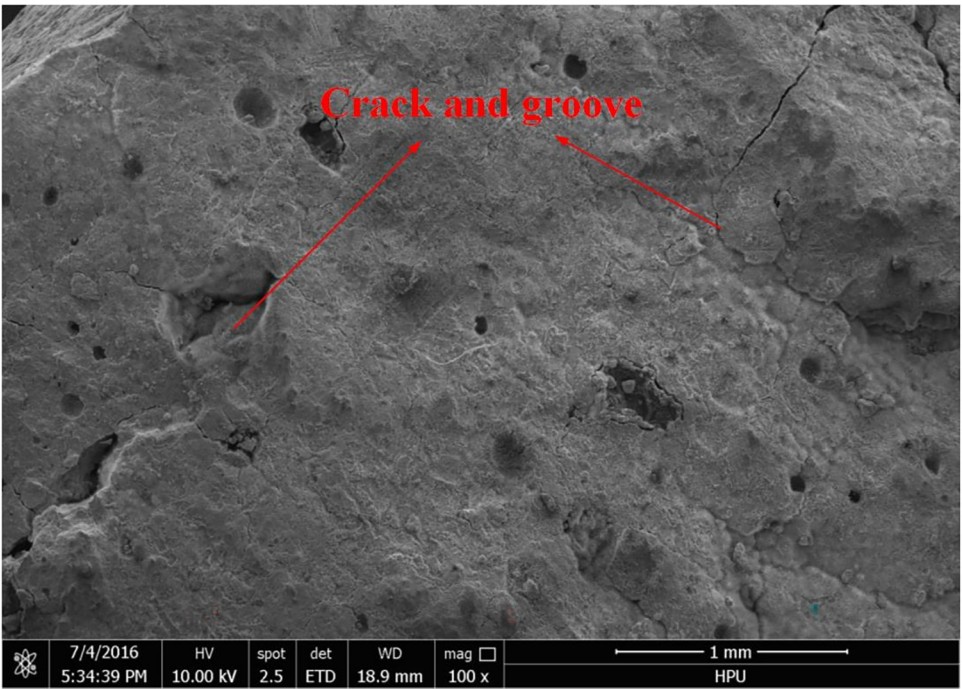

**Fig 7. 28d (×100).**

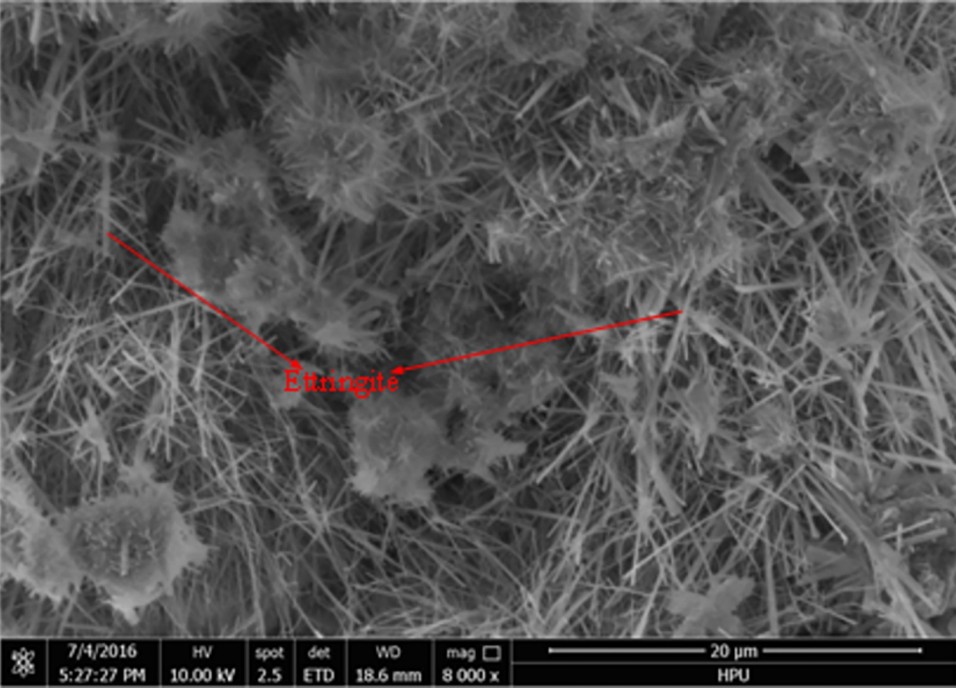

**Fig 8. 3d (×8000).**

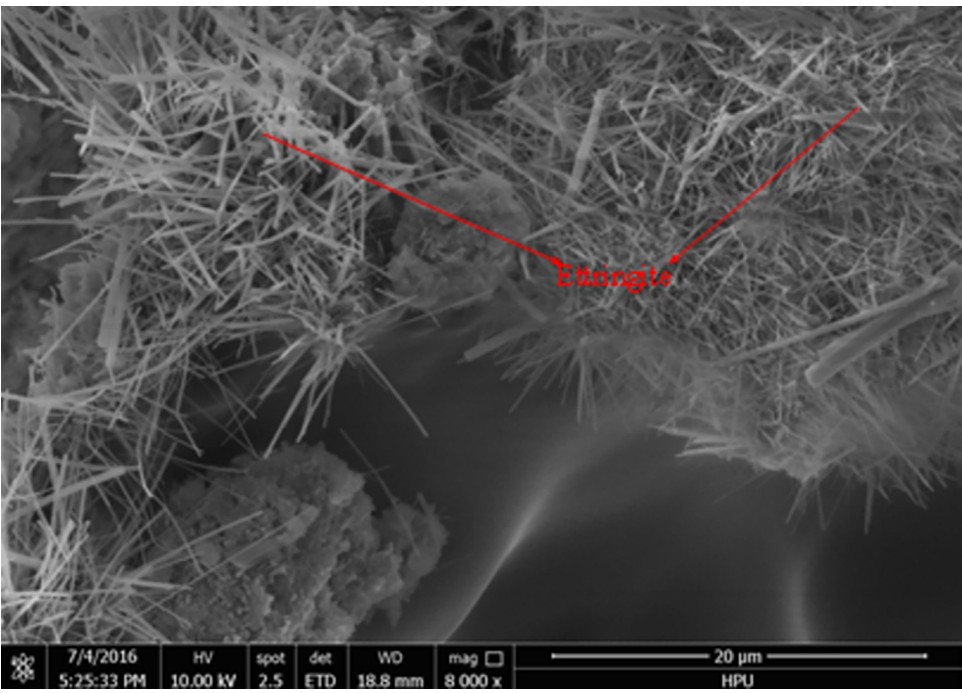

**Fig 9. 28d (×8000).**

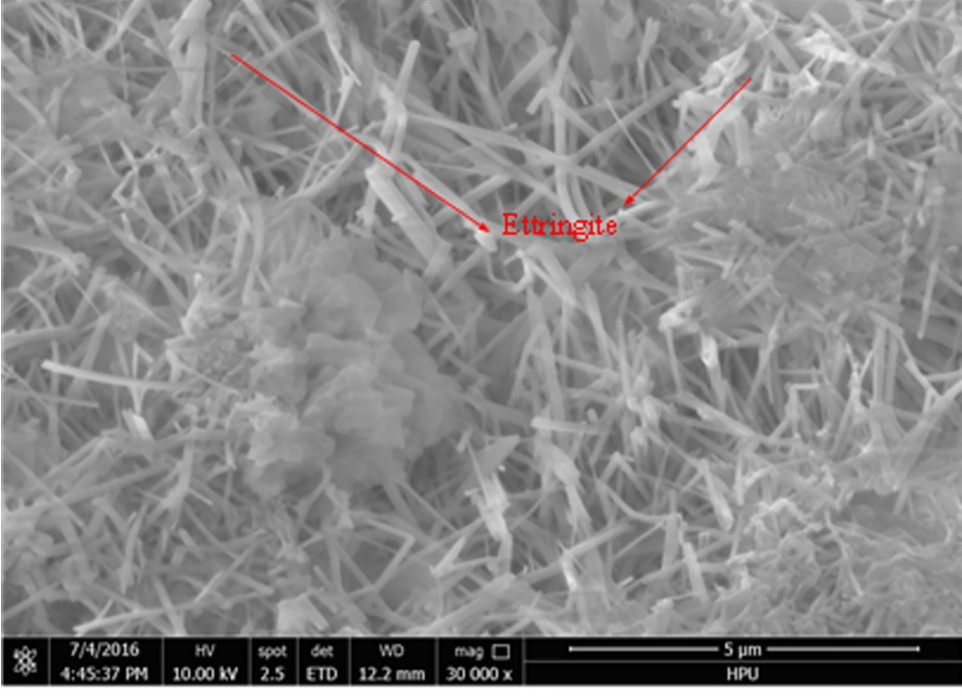

**Fig 10. 3d (×30000).**

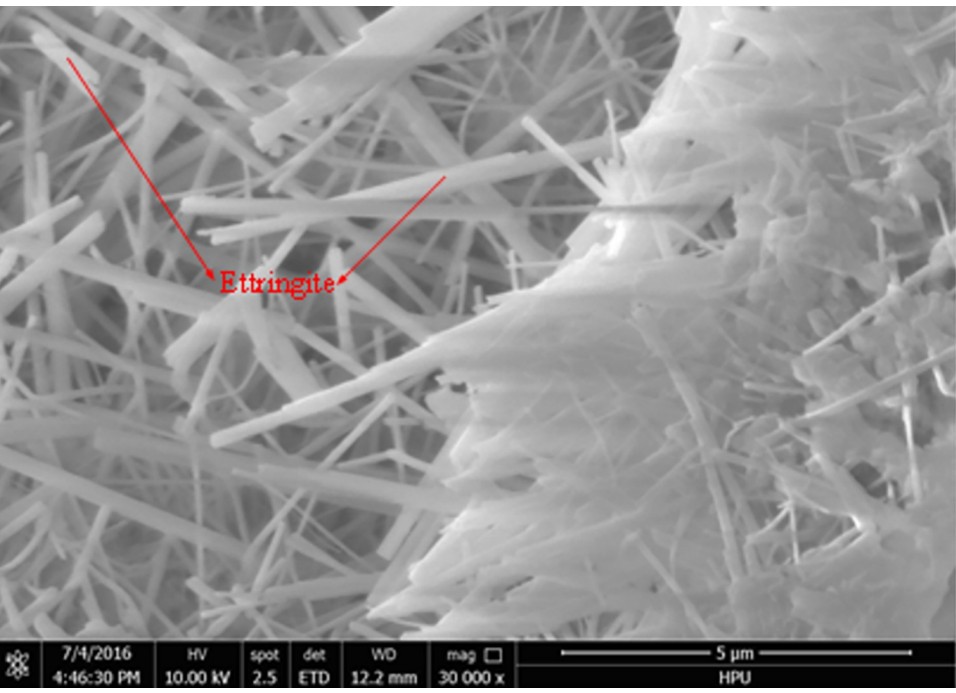

**Fig 11. 28d (×30000).**

aluminates reacted to produce ettringite. These reactions are illustrated in reaction Eqs (1–5).

$$CaO + H_2O \rightarrow Ca(OH)_2 + 15.6 kcal/mol \qquad (1)$$

$$Ca(OH)_2 \rightarrow Ca^{2+} + 2(OH)^{-1} \qquad (2)$$

$$Ca^{2+} + 2(OH)^{-1} + Al_2O_3 \rightarrow \text{calcium aluminate hydrate (CAH)} \qquad (3)$$

$$Ca^{2+} + 2(OH)^{-1} + SiO_2 \rightarrow \text{calcium silicate hydrate (CSH)} \qquad (4)$$

$$Ca^{2+} + 2(OH)^{-1} + Al_2O_3 \rightarrow \text{calcium aluminium silicate hydrate (CASH)} \qquad (5)$$

The state of hydration products was studied to further determine the sealing material microstructure. Figs 8 and 10 shows the porous nature of the material. A small amount of needle-like ettringite was generated due to the presence of sulphate, providing the material with a high uniaxial compressive strength. With the increased aging of the sample, Figs 9 and 11 shows that acicular ettringite crystals were gradually transformed into rod-like structures with the increase in volume, forming a connected network skeleton, which subsequently increased the uniaxial compressive strength. The results of the microstructural analysis showed that the optimum material ratio (as determined by orthogonal tests) resulted in a high uniaxial compressive strength.

## Analysis of the mechanical properties of the sealing material

As shown in Fig 12, the uniaxial compressive strength of the sealing material reached 11.06 MPa, which satisfied the reasonable strength predicted by orthogonal experiments and numerical simulations. After the sample reached its peak strength, the stress decreased slowly, and the curve tended to become smooth with a more extensive deformation range. Meanwhile, the degree of strain softening was relatively low. The sealing material used physical foaming and exhibited an excellent expansion performance due to the presence of internal bubbles. When the load exceeded the maximum value, internal air bubbles were subjected to a constant load and gradually closed, increasing axial deformation. Compressive deformation ensured structural integrity, with the load-bearing capacity not decreasing instantly. Therefore, uniaxial compressive strength and compressive deformation characteristics were improved by optimizing the amounts of additives, achieving optimal results. As shown in Fig 13, with the increase of confining pressure, the compressive strength of the sample increased. The plastic deformation characteristics were more evident when the sample reached its peak. Meanwhile, no evident strain softening was observed, and the gas sealing material showed great plastic deformation.

## Study on sealing length of the gas extraction borehole

The 23302 working face track channel of Yungaishan No. 2 mine was selected for field application. The stress distribution was analyzed theoretically using the drilling cuttings method, and the sealing hole length was determined. The cutting sampling borehole was located at the same level as the gas extraction borehole, which was located in the middle of the adjacent gas extraction borehole. Ten drilling cutting sampling holes were arranged and labeled according to the drilling numbers (1–10#). The drilling distance from the bottom plate was 1.2 m, with a hole

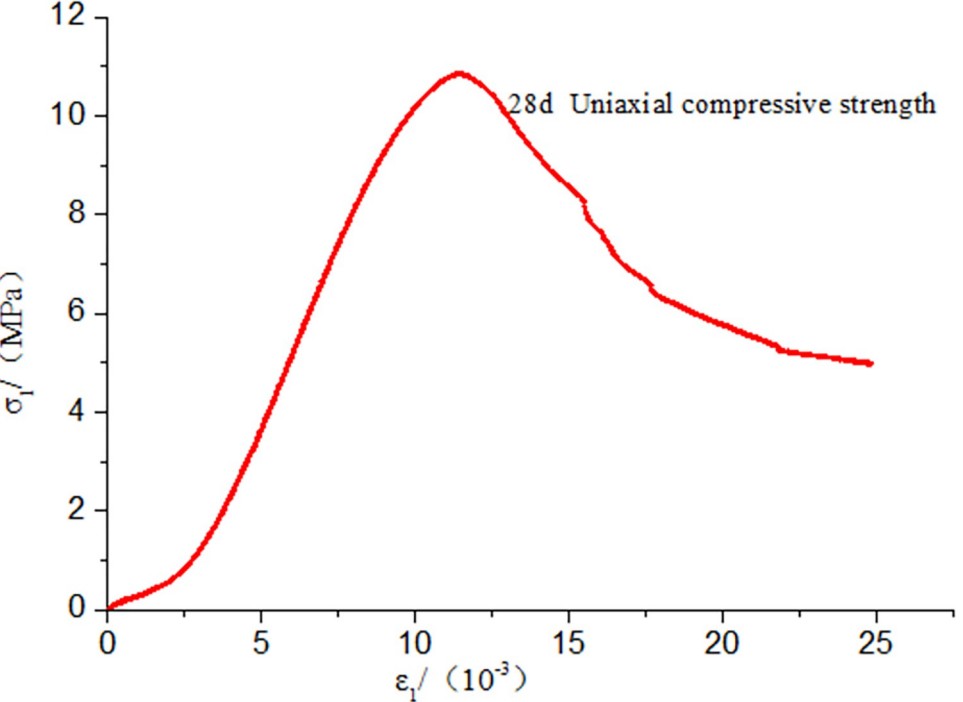

**Fig 12. Uniaxial compressive stress-strain curve for the sealing material.**

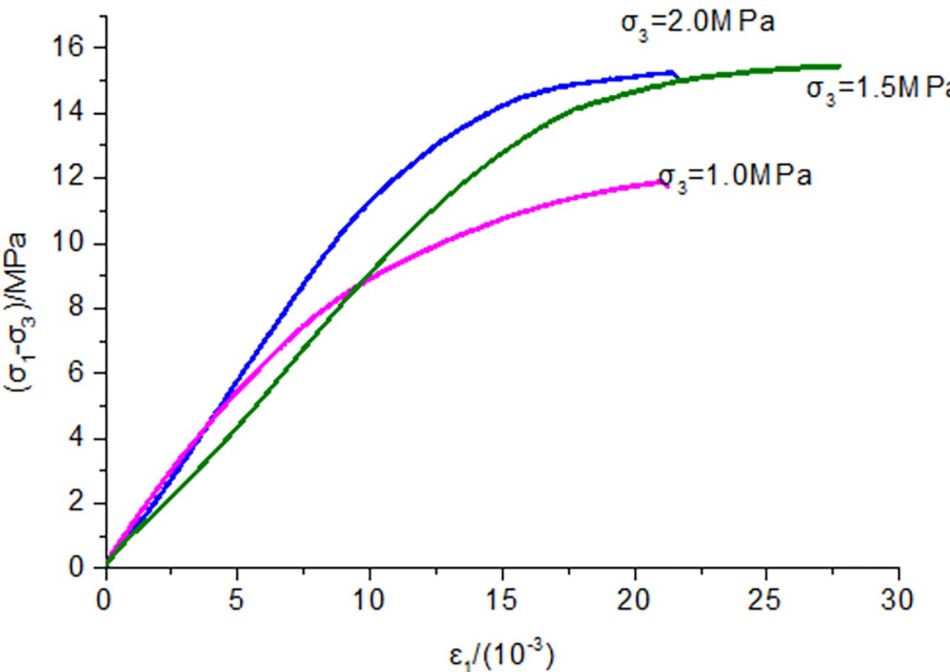

**Fig 13. Triaxial compression stress-strain curve for the sealing material.**

depth of 15 m, hole spacing of 3 m, and data recorded every 1 m. The fitting curve of drilling depth and cuttings quantity is shown in Figs 14 and 15.

As shown in Figs 14 and 15, when the hole depth was within the range of 1–5 m, the cuttings exhibited little change, showing a gradually increasing trend. When the hole depth was increased to 6–8 m, the cuttings increased rapidly, reaching a peak level when the hole depth increased to 9–10 m. This result indicates that the pressure relief zone was within 0–5 m, while the stress increased within 6–8 m. Peak stress appeared at 9 m, and after 9 m, the stress began to decrease gradually. Overall, the reasonable hole sealing depth should exceed the pressure relief zone and be less than the stress peak point, resulting in a reasonable hole sealing depth of 6–9 m for the 23302 roadway in Yungaishan No. 2 mine.

## Field application results

The Yungaishan No. 2 mine was selected to conduct the field testing of the proposed material. The test site was set as the track trough of the 23302 working face. The hole diameter was 84 mm, and the hole spacing was 3 m. The hole-sealing method was adopted for the experiment. Group A used the original cement material with an average drilling depth of 78.8 m. Group B used the novel dual-liquid gas sealing material to seal the boreholes, with an average drilling depth of 85.1 m.

The coal seam drilling experiments consisted of two groups that were 84 mm in diameter. The drilling experiments were designed to assess A) the cement sealing material and B) the novel dual-liquid gas sealing material. Each group contained ten boreholes grouped in two lines and separated by a distance of 3 m. First, the boreholes were drilled, and after the completion of each borehole, gas drainage was started with data monitoring.

For accurate analysis of gas concentration distribution, the frequency of monitoring data generated for each borehole is shown in Fig 16. The curve features are classed as a left 'double

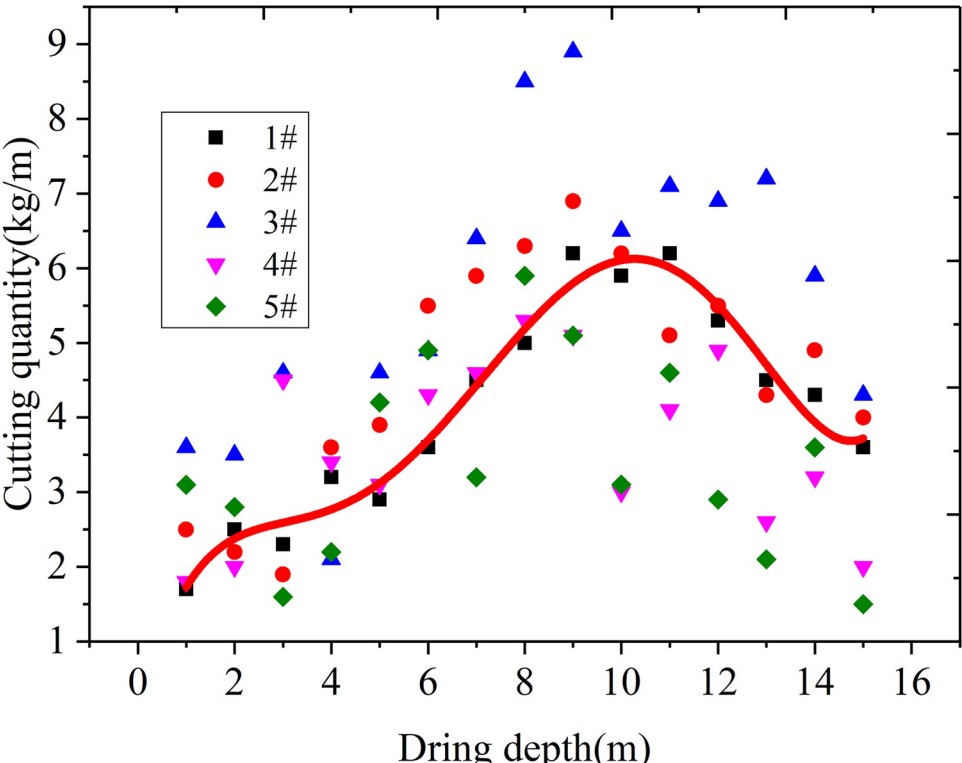

**Fig 14. Relationship between the drilling depth and the cuttings quantity at the 10 selected sampling sites of the 23302 working face 1–5#.**

normal' (Fig 16), with the gas concentration of the cement seal being higher in the earlier stage and decaying rapidly in the later stage. The main reason for this is that when the cement was emplaced, the gelling time became excessively long, and the slurry continued to spread within the cracks, thus resulting in the sealed section not being wholly blocked (some low-density seal areas). Due to this reason, an increase in the leakage channels was observed. The gas concentration was below 30% and accounted for 30.1%. Moreover, more than 80% of gas concentration accounted for only 13.3%. As shown in Fig 17, the curve features are a right 'double normal' type, indicating that the gas extraction concentration was within a high concentration range, with the novel material achieving a good sealing effect. The gas concentration was not less than 30% and accounted for 1.8%. Moreover, more than 80% gas concentration accounted for 30.9%.

To further verify the rationality of the novel sealing material, other mines were chosen for field testing. Two drill fields were selected for the test, with Drill field 1 equipped with 5 drill holes (1#, 2#, 3#, 4#, and 5#) and used high-cost polyurethane as the sealing material. Drill field 2 was also equipped with 5 drill holes (1#, 2#, 3#, 4#, and 5#) and used the novel dual-liquid gas sealing material. The distance between the holes was 3 m, with a hole diameter of 113 mm and a drilling depth of 115 m. The drilling parameters are described in full in Table 4.

As shown in Fig 18, the polyurethane sealing material tended to have low gas concentrations, while the novel sealing material maintained high gas concentrations, with an initial gas concentration of about 40.7% that reduced to 27.6% after attenuation at around 15d. In the prophase, the gas concentration achieved using the novel material was 2.4-fold greater than that of polyurethane and remained 2.1x that of polyurethane in the later stage of the process. Due to differences

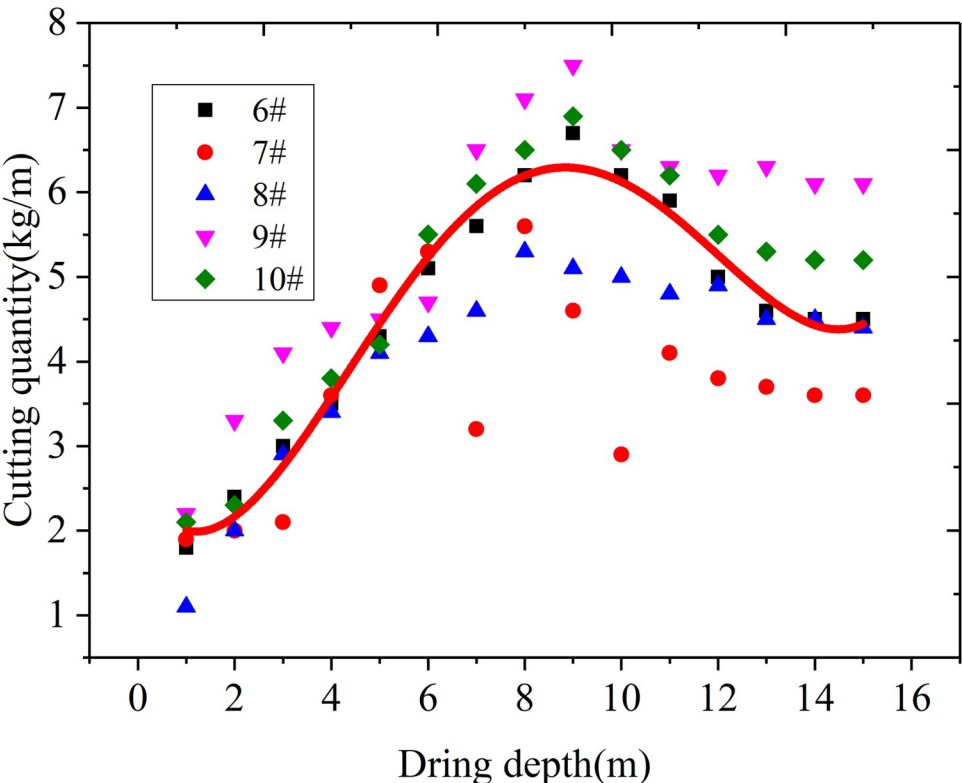

**Fig 15. Relationship between the drilling depth and the cuttings quantity at the 10 selected sampling sites of the 23302 working face 6–10#.**

in gas occurrence in the coal seam, the gas concentration in the second field test was lower than in the first test. This field application verified that the performance of gas extraction using the novel sealing material was higher than that obtained using polyurethane sealing.

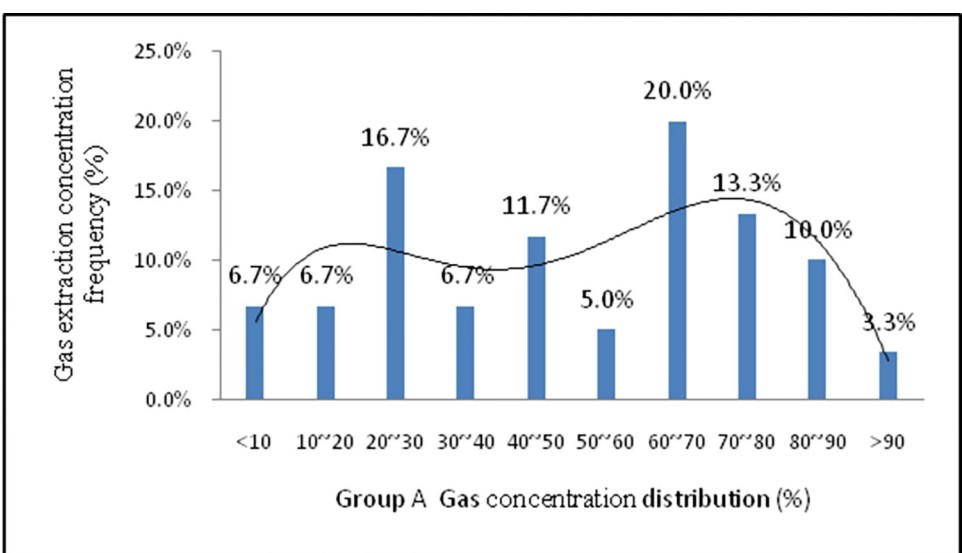

**Fig 16. Gas concentration frequency distribution histogram Group A.**

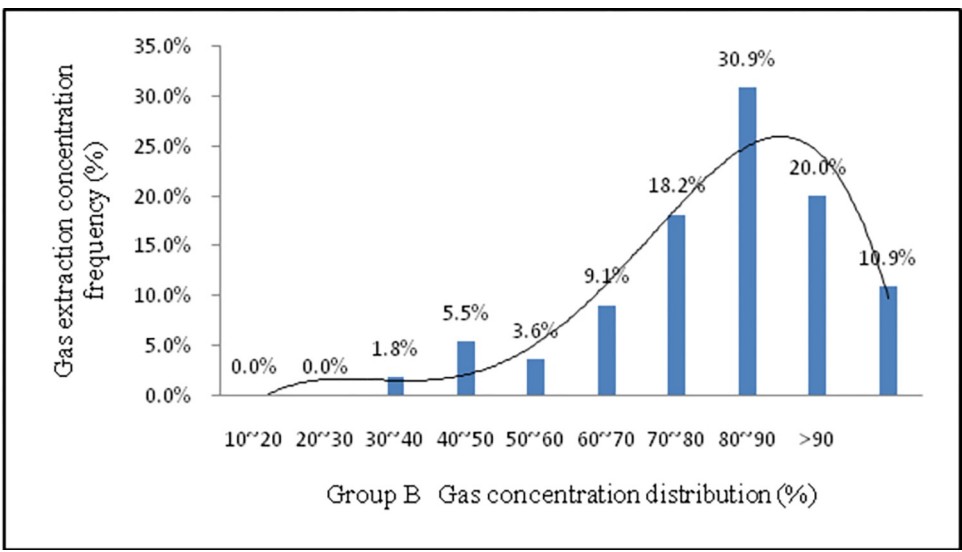

**Fig 17. Gas concentration frequency distribution histogram Group B.**

It can be seen from Fig 19 that, within low gas concentration ranges, the polyurethane sealing material performed much better than the dual-liquid sealing material. When the proportion of gas extraction concentration was less than 10%, accounting for 48.6%, it was nearly 50%. Furthermore, gas extraction concentrations of greater than 50% were not observed during the testing. In comparison, the test data for the novel dual-liquid sealing material showed that gas extraction concentrations of less than 10% only accounted for 17.7%, 2.8x less than that observed for the polyurethane material. When the extraction concentration reached more than 30%, the novel sealing material performed better than polyurethane, with the high concentration of the novel sealing material accounting for a large percentage, indicating that the novel sealing material achieved adequate sealing effects.

## Conclusions

In this paper, a new type of double-liquid gas sealing material was developed. The mechanical properties and microstructure of the proposed material were determined. Furthermore, the

**Table 4. Field test drilling construction parameters.**

| Drilling field | No. | Inclination of borehole/° | Depth of borehole (m) | Diameter of borehole (mm) | Sealing material | The average depth of borehole (m) |
|---|---|---|---|---|---|---|
| 1# Drilling field | 1# | 3 | 115 | 113 | Polyurethane sealing material | 115 |
| | 2# | 4.5 | 115 | 113 | | |
| | 3# | 5 | 115 | 113 | | |
| | 4# | 5 | 115 | 113 | | |
| | 5# | 5 | 115 | 113 | | |
| 2# Drilling field | 1# | 3 | 115 | 113 | Dual-liquid sealing material | 115 |
| | 2# | 5 | 115 | 113 | | |
| | 3# | 5 | 115 | 113 | | |
| | 4# | 5 | 115 | 113 | | |
| | 5# | 5 | 115 | 113 | | |

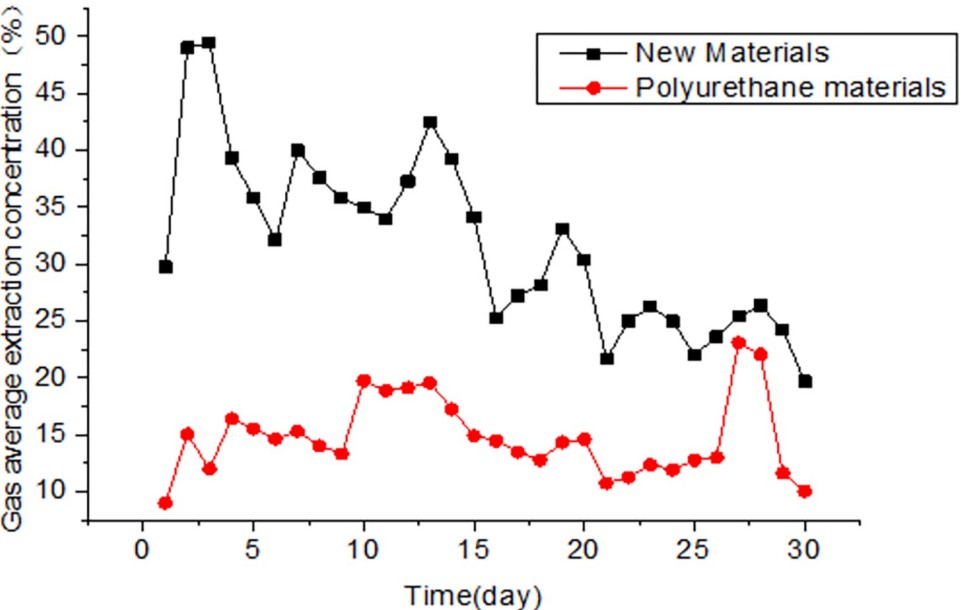

**Fig 18. Comparison of the gas pressure using conventional polyurethane sealing material and the novel dual-liquid sealing material.**

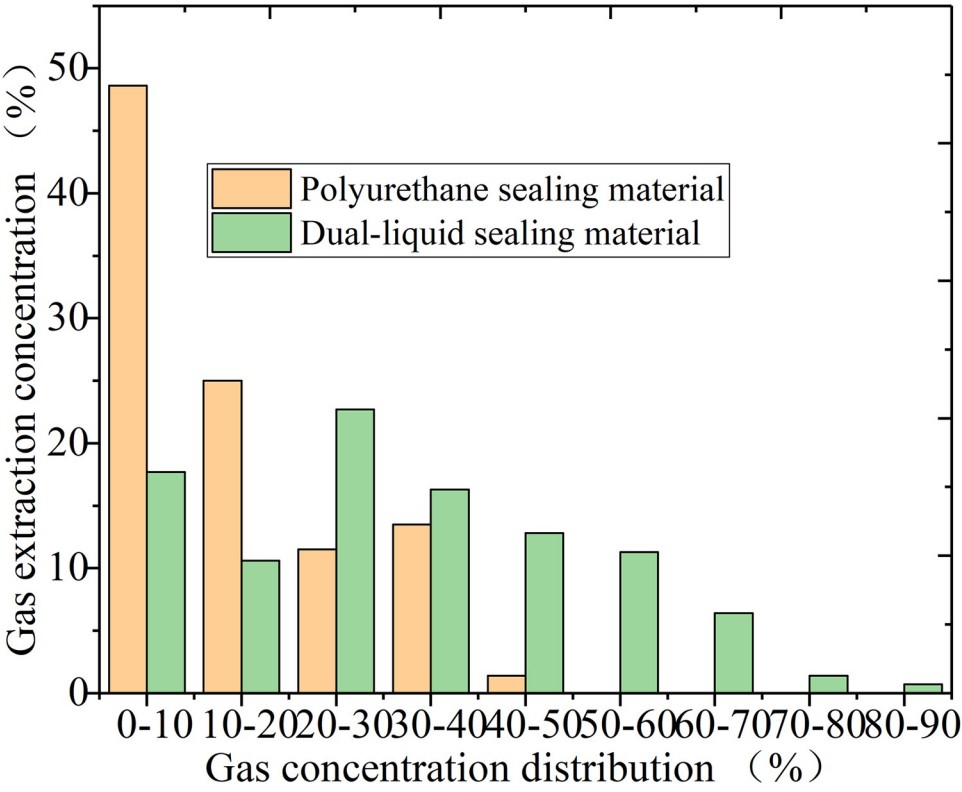

**Fig 19. Gas concentration frequency statistics comparison chart.**

performance of the new dual-liquid sealing material was verified using field applications, and the following conclusions were obtained.

1. The optimal compositions of a compound accelerant (0.3%), a compound retarder (0.5%), an expansion agent (0.03%), and a water-cement ratio (1.2:1.0) were determined using orthogonal tests. The optimal composition of the new dual-liquid sealing material was determined. Under these conditions, the gelation time was 9 min. The final setting time was 23 min. The sample density was 1.25 $g/m^3$, while the fluidity was 336 mm. The expansion ratio was 6.8%, and the 28-d compressive strength was 11.06 MPa. Theoretically, the sealing material's properties met the requirements for field application.

2. Microstructure analysis showed that, with the increase in sample age, the formation of hydration products increased significantly with the generation of denser networks. The structure of gelled ettringite gradually changed from needles to thick bars, forming a network skeleton. The results showed that the novel sealing material exhibited high compressive strength, good plastic deformation characteristics, and a large deformation capacity. The axial strain was shown to be significant, and effective compression performance was guaranteed.

3. Field application showed that after the novel material sealed the borehole, the gas drainage concentration improved, and the drainage cycle was prolonged. The concentration of gas extraction was typically higher than 30%, and the drainage effect improved significantly.

## Supporting information

**S1 Data.**
(XLSX)

## Acknowledgments

We thank Mr. Zuo from the China University of Mining and Technology, Beijing, for his expertise and kind assistance in our studies. Furthermore, the authors would like to thank Professor Li and Professor Xu for their support.

We would like to thank MogoEdit (https://www.mogoedit.com) for its English editing during the preparation of this manuscript.

## Author Contributions

**Data curation:** Zijie Hong, Lei Xu.

**Formal analysis:** Zhenhua Li, Lei Xu.

**Funding acquisition:** Zhenhua Li.

**Project administration:** Zijie Hong.

**Resources:** Zhenhua Li.

**Supervision:** Jianping Zuo.

**Writing – original draft:** Zijie Hong.

**Writing – review & editing:** Jianping Zuo.

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
