## [Decision Letter · Decision Letter 0]

5 Oct 2022

PONE-D-22-23213Experimental characterization and field application of a novel dual-liquid gas leakage material: mechanical properties and microscopic hydration mechanismPLOS ONE

Dear Dr. Hong,

Thank you for submitting your manuscript to PLOS ONE. After careful consideration, we feel that it has merit but does not fully meet PLOS ONE’s publication criteria as it currently stands. Therefore, we invite you to submit a revised version of the manuscript that addresses the points raised during the review process.

We look forward to receiving your revised manuscript.

Kind regards,

Jianguo Wang, PhD

Academic Editor

PLOS ONE

Journal Requirements:

"NO authors have competing interests"

Reviewers' comments:

Reviewer's Responses to Questions

**Comments to the Author**

1. Is the manuscript technically sound, and do the data support the conclusions?

Reviewer #1: Yes

Reviewer #2: Yes

2. Has the statistical analysis been performed appropriately and rigorously? 

Reviewer #1: Yes

Reviewer #2: Yes

3. Have the authors made all data underlying the findings in their manuscript fully available?

Reviewer #1: Yes

Reviewer #2: Yes

4. Is the manuscript presented in an intelligible fashion and written in standard English?

Reviewer #1: Yes

Reviewer #2: Yes

5. Review Comments to the Author

Reviewer #1: The manuscript analyzed the mechanism of gas drainage leakage, developing a novel dual-liquid gas sealing material and determining its mechanical properties and hydration mechanism. Meanwhile, the performance of the novel dual-liquid sealing material was verified via field application. It was found that the new sealing material can improve the gas drainage performance by improving the compressive strength and reducing the damage. However, the paper suffers for some serious limits:

Major comments

1. From the introduction section of the manuscript, the authors have introduced the relevant background of gas drainage, and have also summarized the relevant research content of some scholars. However, this section still requires a more detailed summary of the existing research progress and problems to be solved (more than lines 61-64). The reviewer tends to recommend that the authors should complete the research background introduction, point out the deficiencies in the related fields, and state the improvement of the proposed new methods/new materials for the deficiencies.

2. In the mechanism analysis section (from line 70), according to the reviewer, the authors have introduced the present research progress on gas drainage leakage. However, it is not very clear why the authors present this section independently of the Introduction section, which may confuse readers whether the content of this section is the experimental research carried out by the authors or the review of previous studies. In particular, the cited literature seems relatively few, which is not very convincing to support the authors’ view. In addition, as an independent section with a short length, it is suggested to merge with the Introduction section. Also, Figures 1 and 2 don’t seem very clear and need to be replaced with a higher picture quality.

3. In the experimental study section, before introducing the composition of the materials, please explain the reasons for selecting these materials, which may be due to the consideration of strength, plastic deformation or other factors. And it is encouraged to include pictures of relevant materials after description instead of only few tables.

4. The reviewer thinks that the advanced nature of the proposed material method should mainly focus on the comparison with the performance of traditional materials or the most commonly used material methods. However, the comparison part in the paper seems to only be the comparison of gas extraction. Is it necessary to compare the mechanical properties at laboratory scale or the mechanism at a microscopic scale?

Other specific comments

5. There are too many chapters in the results and discussion section, resulting in a small amount of content in each chapter. It is suggested to reduce the chapter amounts a bit.

6. Although there is “microscopic hydration mechanism” words in the title, the relevant text introduction is very little (only more than 20 lines of concentrated introduction), Should the authors need to consider modifying the title or adding relevant content research?

For these reasons, I suggest that this work needs a major revision.

Reviewer #2: In this study, the complexities of gas occurrence and migration was investigated, and the mechanism responsible for the poor sealing effect of traditional sealing materials was established. The novel dual-liquid gas sealing material reduced damage to the rock surrounding the borehole, significantly improving the gas drainage performance. The topic of this study is very meaningful and innovative. It is suggested the manuscript can be improved before publication. And there are several problems with this paper:

(1) Some references need to be updated so that authors can keep up with the latest research findings.

(2) The principle of gas leakage is well explained in Fig.1, however the author's description is not detailed enough. Please refer to the arrow description in the picture.

(3) In “Gypsum and lime ratio determination” section, for the mechanical properties of the material, why only 3 days mechanical parameters are given in Table 1 and Fig. 3.

(4) When the author explained the material mechanism, the author mentioned ettringite and other substances. Why didn't the author mark them in Fig. 4, for example, Fig 4 (a1).

(5) In “Analysis of the mechanical properties of sealing materials” section, the authors have done the test of sealing material, but did not analyze the effect of confining pressure on mechanical properties. Please add it.

(6) The conclusions should be simplified and summarized, please revise it.

(7) Some sentences have spelling mistakes, please correct carefully.

6. PLOS authors have the option to publish the peer review history of their article (what does this mean?). If published, this will include your full peer review and any attached files.

Reviewer #1: No

Reviewer #2: No

---

## [Author Response · Author response to Decision Letter 0]

27 Nov 2022

Dear Prof./Dr. Jianguo Wang Editor-in-Chief:

Thank you very much for your letter. All authors would like to take this opportunity to express our gratitude to your valuable comments and suggestions on (Ref: PONE-D-22-23213). Paper title: “Experimental characterization and field application of a novel dual-liquid gas leakage material: mechanical properties and microscopic hydration mechanism”.

According to your suggestions, we have carefully modified the manuscript and made it easily to be understood. We hope the revised manuscript can meet these requests and be accepted. The revised manuscript is attached here, and the questions are answered one by one in the enclosed files.

I am looking forward to receiving your decision as soon as possible.

With my best regards.

Zi-jie Hong, Jian-ping Zuo, Zhen-hua Li, Lei Xu

 

Responds to the academic editor’s comments:

Comment 1: Please ensure that your manuscript meets PLOS ONE's style requirements, including those for file naming.

Response: Thank you for your valuable advice. The authors completed the submission in strict accordance with the requirements of the journal.

Comment 2: We note that the grant information you provided in the ‘Funding Information’ and ‘Financial Disclosure’ sections do not match. When you resubmit, please ensure that you provide the correct grant numbers for the awards you received for your study in the ‘Funding Information’ section.

Response: Thank you for your valuable advice. The authors have ensured that we provide the correct grant numbers for the awards we received for our study in the ‘Funding Information’ section.

Comment 3: Thank you for stating the following in your Competing Interests section: 

"NO authors have competing interests"

Response: Thank you for your valuable advice. The authors have declared that no competing interests exist.

Comment 4: We note that you have stated that you will provide repository information for your data at acceptance. Should your manuscript be accepted for publication, we will hold it until you provide the relevant accession numbers or DOIs necessary to access your data. If you wish to make changes to your Data Availability statement, please describe these changes in your cover letter and we will update your Data Availability statement to reflect the information you provide.

Response: Thank you for your valuable advice. We will provide repository information for your data at acceptance.

Data Availability Statement

jie, hong, Henan Polytechnic University, https://orcid.org/0000-0002-1946-6754

hongzijie2010@126.com

Publication date: Not available

Publisher: Dryad

Research Facility: Henan Polytechnic University

https://doi.org/10.5061/dryad.5x69p8d79

Citation

jie, hong (2022), Data Availability Statement, Dryad, Dataset, https://doi.org/10.5061/dryad.5x69p8d79

#1 Responds to the Reviewer’s comments:

Comment 1: From the introduction section of the manuscript, the authors have introduced the relevant background of gas drainage, and have also summarized the relevant research content of some scholars. However, this section still requires a more detailed summary of the existing research progress and problems to be solved (more than lines 61-64). The reviewer tends to recommend that the authors should complete the research background introduction, point out the deficiencies in the related fields, and state the improvement of the proposed new methods/new materials for the deficiencies.

Response: Thank you for your valuable advice. The authors tend to revisions as suggested by the reviewer, including the deficiencies in the related fields, and state the improvement of the proposed new methods/new materials for the deficiencies. And the revised portion is shown in this paper with red color.

Comment 2: In the mechanism analysis section (from line 70), according to the reviewer, the authors have introduced the present research progress on gas drainage leakage. However, it is not very clear why the authors present this section independently of the Introduction section, which may confuse readers whether the content of this section is the experimental research carried out by the authors or the review of previous studies. In particular, the cited literature seems relatively few, which is not very convincing to support the authors’ view. In addition, as an independent section with a short length, it is suggested to merge with the Introduction section. Also, Figures 1 and 2 don’t seem very clear and need to be replaced with a higher picture quality.

Response: Thank you for your valuable and thoughtful comments. This is a constructive suggestion by the reviewers. The authors took the suggestion, and we eventually merged it with Introduction section.

Comment 3: In the experimental study section, before introducing the composition of the materials, please explain the reasons for selecting these materials, which may be due to the consideration of strength, plastic deformation or other factors. And it is encouraged to include pictures of relevant materials after description instead of only few tables.

Response: Thank you very much to point out the important problem. According to the comments, and the revised portion is shown in this paper with red color. The revisions are also shown as follows:

Sulphoaluminate cement clinker has the advantages of strong early strength, high strength, good erosion resistance, impermeability and frost resistance. Sulfur building gypsum and quicklime can improve the compressive strength of cement materials. The retarder can be adsorbed on the surface of the cement, slowing down the cement hydration process, thus extending the slurry loss time. The thickener can increase the viscosity of the slurry, improve the anti-segregation performance of the slurry, and also improve the strength of the consolidated body.

(a) Sulphoaluminate cement clinker (b) Sulfur building gypsum and quicklime

(c) The retarder (d) The thickener

Fig 1 Materials used in laboratory

Comment 4: The reviewer thinks that the advanced nature of the proposed material method should mainly focus on the comparison with the performance of traditional materials or the most commonly used material methods. However, the comparison part in the paper seems to only be the comparison of gas extraction. Is it necessary to compare the mechanical properties at laboratory scale or the mechanism at a microscopic scale?

Response: Thank you very much to point out the important problem. The main research background of this paper is based on the serious gas leakage problem in coal seam gas extraction, and reasonable sealing material is the key parameter to reduce gas leakage. Therefore, this paper mainly focuses on the influence of the new materials on the gas extraction effect, and compares the effect of the traditional materials on the gas extraction effect.

Comment 5: There are too many chapters in the results and discussion section, resulting in a small amount of content in each chapter. It is suggested to reduce the chapter amounts a bit.

Response: Thank you for your valuable and thoughtful comments. According to the comments, the authors have reduced the chapter amounts a bit.

Comment 6: Although there is “microscopic hydration mechanism” words in the title, the relevant text introduction is very little (only more than 20 lines of concentrated introduction), Should the authors need to consider modifying the title or adding relevant content research?

Response: Thank you for your valuable and thoughtful comments. According to the comments, the authors carefully considered the relationship of the paper, and the authors have changed the title to Analysis of the Micromorphology.

#2 Responds to the Reviewer’s comments:

Comment 1: Some references need to be updated so that authors can keep up with the latest research findings.

Response: Thank you for your valuable and thoughtful comments. According to the comments, the authors have revised the references.

Comment 2: The principle of gas leakage is well explained in Fig.1, however the author's description is not detailed enough. Please refer to the arrow description in the picture.

Response: Thank you for your valuable and thoughtful comments. According to the comments, and combined with the comments given by Reviewer’s 2 comments, the author integrates the chapter with the Introduction section.

Comment 3: In “Gypsum and lime ratio determination” section, for the mechanical properties of the material, why only 3 days mechanical parameters are given in Table 1 and Fig. 3.

Response: Thank you for your valuable and thoughtful comments. Table 1 and Fig 3 are the data obtained to study the optimal ratio of gypsum to lime. The change rule of the previous 3 days can be used to obtain the optimal mixing ratio. Therefore, only 3 days mechanical parameters are given in Table 1 and Fig. 3.

Comment 4: When the author explained the material mechanism, the author mentioned ettringite and other substances. Why didn't the author mark them in Fig. 4, for example, Fig 4 (a1).

Response: Thank you for your valuable and thoughtful comments. According to the comments, the authors have marked ettringite in the corresponding position in Fig. 4.

(b) 3d (×8000) (b1) 28d (×8000)

(c) 3d (×30000) (c1) 28d (×30000)

Fig 3. Scanning electron micrograph of the sealing material at different stages of ageing

Comment 5: In “Analysis of the mechanical properties of sealing materials” section, the authors have done the test of sealing material, but did not analyze the effect of confining pressure on mechanical properties. Please add it.

Response: Thank you for your valuable and thoughtful comments. According to the comments, the revised portion is shown in this paper with red color. The revisions are also shown as follows:

As shown in Fig. 6, with the increase of confining pressure, the compressive strength of the sample increases. The plastic deformation characteristics are more obvious, when the sample reached the peak, no obvious strain softening was observed, the gas sealing material shows great plastic deformation. 

Comment 6: The conclusions should be simplified and summarized, please revise it.

Response: Thank you for your valuable and thoughtful comments. According to the comments, the authors have revised the conclusions, the revised portion is shown in this paper with red color.

Comment 7: Some sentences have spelling mistakes, please correct carefully.

Response: Thank you for your valuable and thoughtful comments. According to the comments, the authors have carefully revised the sentences.

---

## [Decision Letter · Decision Letter 1]

13 Dec 2022

PONE-D-22-23213R1Experimental characterization and field application of a novel dual-liquid gas leakage material: mechanical properties and microscopic hydration mechanismPLOS ONE

Dear Dr. Hong,

Thank you for submitting your manuscript to PLOS ONE. After careful consideration, we feel that it has merit but does not fully meet PLOS ONE’s publication criteria as it currently stands. Therefore, we invite you to submit a revised version of the manuscript that addresses the points raised during the review process.

ACADEMIC EDITOR:Minor revision is still necessary. ============================== Please submit your revised manuscript by Jan 27 2023 11:59PM. If you will need more time than this to complete your revisions, please reply to this message or contact the journal office at plosone@plos.org. Please include the following items when submitting your revised manuscript:A rebuttal letter that responds to each point raised by the academic editor and reviewer(s). You should upload this letter as a separate file labeled 'Response to Reviewers'.A marked-up copy of your manuscript that highlights changes made to the original version. You should upload this as a separate file labeled 'Revised Manuscript with Track Changes'.An unmarked version of your revised paper without tracked changes. You should upload this as a separate file labeled 'Manuscript'.If applicable, we recommend that you deposit your laboratory protocols in protocols.io to enhance the reproducibility of your results. Protocols.io assigns your protocol its own identifier (DOI) so that it can be cited independently in the future. For instructions see: https://journals.plos.org/plosone/s/submission-guidelines#loc-laboratory-protocols. Additionally, PLOS ONE offers an option for publishing peer-reviewed Lab Protocol articles, which describe protocols hosted on protocols.io. Read more information on sharing protocols at https://plos.org/protocols?utm_medium=editorial-email&utm_source=authorletters&utm_campaign=protocols.

We look forward to receiving your revised manuscript.

Kind regards,

Jianguo Wang, PhD

Academic Editor

PLOS ONE

Journal Requirements:

Reviewers' comments:

Reviewer's Responses to Questions

**Comments to the Author**

1. If the authors have adequately addressed your comments raised in a previous round of review and you feel that this manuscript is now acceptable for publication, you may indicate that here to bypass the “Comments to the Author” section, enter your conflict of interest statement in the “Confidential to Editor” section, and submit your "Accept" recommendation.

Reviewer #1: (No Response)

Reviewer #2: All comments have been addressed

2. Is the manuscript technically sound, and do the data support the conclusions?

Reviewer #1: Yes

Reviewer #2: Yes

3. Has the statistical analysis been performed appropriately and rigorously? 

Reviewer #1: Yes

Reviewer #2: Yes

4. Have the authors made all data underlying the findings in their manuscript fully available?

Reviewer #1: Yes

Reviewer #2: Yes

5. Is the manuscript presented in an intelligible fashion and written in standard English?

Reviewer #1: Yes

Reviewer #2: Yes

6. Review Comments to the Author

Reviewer #1: On the basis of the previous manuscript, the authors made a detailed supplement to the background section, and added and corrected the relevant research content, respectively. The overall structure of the article has been greatly improved. However, there are still some problems, and the authors need a minor revision.

1. As an experimental research on gas drainage, is it necessary to introduce a simple parameter introduction (in the form of pictures or text descriptions) to the overall storage model or prototype site? It seems that there is no relevant content in the draft.

2. As a physical experiment, the verification of accuracy (or how to avoid experimental errors) is a very important part. Should the authors add the verification content into the manuscript?

3. The picture quality in the manuscript is so poor that the details of the picture and the labels on the picture are hard to see (such as SEM pictures). However, it is clearer in the response section on the contrary, the author should attach pictures with higher-quality versions in the text.

4. It is recommended to polish the overall text to make the expression more reasonable.

5. If the pictures are listed separately, it is recommended to add the titles below the pictures instead of leaving the titles in the text.

Reviewer #2: The manuscript has been changed according to my comments, it meets the dual publication, research ethics, or publication ethics. it could be accepted in the state.

7. PLOS authors have the option to publish the peer review history of their article (what does this mean?). If published, this will include your full peer review and any attached files.

Reviewer #1: No

Reviewer #2: No

---

## [Author Response · Author response to Decision Letter 1]

26 Jan 2023

Dear Prof./Dr. Jianguo Wang Editor-in-Chief:

Thank you very much for your letter. All authors would like to take this opportunity to express our gratitude to your valuable comments and suggestions on (Ref: PONE-D-22-23213). Paper title: “Experimental characterization and field application of a novel dual-liquid gas leakage material: mechanical properties and microscopic hydration mechanism”.

According to your suggestions, we have carefully modified the manuscript and made it easily to be understood. We hope the revised manuscript can meet these requests and be accepted. The revised manuscript is attached here, and the questions are answered one by one in the enclosed files.

I am looking forward to receiving your decision as soon as possible.

With my best regards.

Zi-jie Hong, Jian-ping Zuo, Zhen-hua Li, Lei Xu

 

Responds to the academic editor’s comments:

Comment 1: As an experimental research on gas drainage, is it necessary to introduce a simple parameter introduction (in the form of pictures or text descriptions) to the overall storage model or prototype site? It seems that there is no relevant content in the draft.

Response: Thank you for your valuable advice. According to the comment, the author adds the field parameter.

Comment 2: As a physical experiment, the verification of accuracy (or how to avoid experimental errors) is a very important part. Should the authors add the verification content into the manuscript?

Response: Thank you for your valuable advice. The results obtained by the author are focused on the mechanical properties of the sealing materials, which can be used to meet the requirements of field tests. According to the requirements, the author completed the field test, and achieved good results. The correctness of the experiment is verified.

Comment 3: The picture quality in the manuscript is so poor that the details of the picture and the labels on the picture are hard to see (such as SEM pictures). However, it is clearer in the response section on the contrary, the author should attach pictures with higher-quality versions in the text.

Response: Thank you for your valuable advice. The problem of picture quality may be due to the system's processing of pictures. All pictures should be processed by the system when submitting, otherwise the submission cannot be completed. Meanwhile, according to the comment, the author revised some of the pictures.

Comment 4: It is recommended to polish the overall text to make the expression more reasonable.

Response: Thank you for your valuable and thoughtful comments. The author used a firm called MogoEdit to complete the revision. The proof of modification is as follows:

Comment 5: If the pictures are listed separately, it is recommended to add the titles below the pictures instead of leaving the titles in the text.

Response: Thank you for your valuable comments. The journal required that the picture and the title should be separated, thus the author treated it according to the author guidance.

---

## [Editor Report · Decision Letter 2]

3 Feb 2023

PONE-D-22-23213R2Experimental characterization and field application of a novel dual-liquid gas leakage material: mechanical properties and microscopic hydration mechanismPLOS ONE

Dear Dr. Hong,

Thank you for submitting your manuscript to PLOS ONE. After careful consideration, we feel that it has merit but does not fully meet PLOS ONE’s publication criteria as it currently stands. Therefore, we invite you to submit a revised version of the manuscript that addresses the points raised during the review process.

ACADEMIC EDITOR: Please insert comments here and delete this placeholder text when finished. Be sure to:Your manuscript almost reaches its final stage. I checked your manuscript and found the necessary improvements as follows: 1, figures 2 and 3: sample age is an unclear terminology. Figures 9 and 10: the number in X-axis is together. They should be separated for each range.  2, Figure 6(b): Dring depth?  Is it 'Drilling depth'?Both English and presentations should be furtehr improved.==============================

We look forward to receiving your revised manuscript.

Kind regards,

Jianguo Wang, PhD

Academic Editor

PLOS ONE
---

## [Author Response · Author response to Decision Letter 2]

20 Mar 2023

Dear Prof./Dr. Jianguo Wang Editor-in-Chief:

Thank you very much for your letter. All authors would like to take this opportunity to express our gratitude to your valuable comments and suggestions on (Ref: PONE-D-22-23213). Paper title: “Experimental characterization and field application of a novel dual-liquid gas leakage material: mechanical properties and microscopic hydration mechanism”.

According to your suggestions, we have carefully modified the manuscript and made it easily to be understood. We hope the revised manuscript can meet these requests and be accepted. The revised manuscript is attached here, and the questions are answered one by one in the enclosed files.

I am looking forward to receiving your decision as soon as possible.

With my best regards.

Zi-jie Hong, Jian-ping Zuo, Zhen-hua Li, Lei Xu

 

First revision

Responds to the academic editor’s comments:

Comment 1: Please ensure that your manuscript meets PLOS ONE's style requirements, including those for file naming.

Response: Thank you for your valuable advice. The authors completed the submission in strict accordance with the requirements of the journal.

Comment 2: We note that the grant information you provided in the ‘Funding Information’ and ‘Financial Disclosure’ sections do not match. When you resubmit, please ensure that you provide the correct grant numbers for the awards you received for your study in the ‘Funding Information’ section.

Response: Thank you for your valuable advice. The authors have ensured that we provide the correct grant numbers for the awards we received for our study in the ‘Funding Information’ section.

Comment 3: Thank you for stating the following in your Competing Interests section: 

"NO authors have competing interests"

Response: Thank you for your valuable advice. The authors have declared that no competing interests exist.

Comment 4: We note that you have stated that you will provide repository information for your data at acceptance. Should your manuscript be accepted for publication, we will hold it until you provide the relevant accession numbers or DOIs necessary to access your data. If you wish to make changes to your Data Availability statement, please describe these changes in your cover letter and we will update your Data Availability statement to reflect the information you provide.

Response: Thank you for your valuable advice. We will provide repository information for your data at acceptance.

Data Availability Statement

jie, hong, Henan Polytechnic University, https://orcid.org/0000-0002-1946-6754

hongzijie2010@126.com

Publication date: Not available

Publisher: Dryad

Research Facility: Henan Polytechnic University

https://doi.org/10.5061/dryad.5x69p8d79

Citation

jie, hong (2022), Data Availability Statement, Dryad, Dataset, https://doi.org/10.5061/dryad.5x69p8d79

#1 Responds to the Reviewer’s comments:

Comment 1: From the introduction section of the manuscript, the authors have introduced the relevant background of gas drainage, and have also summarized the relevant research content of some scholars. However, this section still requires a more detailed summary of the existing research progress and problems to be solved (more than lines 61-64). The reviewer tends to recommend that the authors should complete the research background introduction, point out the deficiencies in the related fields, and state the improvement of the proposed new methods/new materials for the deficiencies.

Response: Thank you for your valuable advice. The authors tend to revisions as suggested by the reviewer, including the deficiencies in the related fields, and state the improvement of the proposed new methods/new materials for the deficiencies. And the revised portion is shown in this paper with red color.

Comment 2: In the mechanism analysis section (from line 70), according to the reviewer, the authors have introduced the present research progress on gas drainage leakage. However, it is not very clear why the authors present this section independently of the Introduction section, which may confuse readers whether the content of this section is the experimental research carried out by the authors or the review of previous studies. In particular, the cited literature seems relatively few, which is not very convincing to support the authors’ view. In addition, as an independent section with a short length, it is suggested to merge with the Introduction section. Also, Figures 1 and 2 don’t seem very clear and need to be replaced with a higher picture quality.

Response: Thank you for your valuable and thoughtful comments. This is a constructive suggestion by the reviewers. The authors took the suggestion, and we eventually merged it with Introduction section.

Comment 3: In the experimental study section, before introducing the composition of the materials, please explain the reasons for selecting these materials, which may be due to the consideration of strength, plastic deformation or other factors. And it is encouraged to include pictures of relevant materials after description instead of only few tables.

Response: Thank you very much to point out the important problem. According to the comments, and the revised portion is shown in this paper with red color. The revisions are also shown as follows:

Sulphoaluminate cement clinker has the advantages of strong early strength, high strength, good erosion resistance, impermeability and frost resistance. Sulfur building gypsum and quicklime can improve the compressive strength of cement materials. The retarder can be adsorbed on the surface of the cement, slowing down the cement hydration process, thus extending the slurry loss time. The thickener can increase the viscosity of the slurry, improve the anti-segregation performance of the slurry, and also improve the strength of the consolidated body.

(a) Sulphoaluminate cement clinker (b) Sulfur building gypsum and quicklime

(c) The retarder (d) The thickener

Fig 1 Materials used in laboratory

Comment 4: The reviewer thinks that the advanced nature of the proposed material method should mainly focus on the comparison with the performance of traditional materials or the most commonly used material methods. However, the comparison part in the paper seems to only be the comparison of gas extraction. Is it necessary to compare the mechanical properties at laboratory scale or the mechanism at a microscopic scale?

Response: Thank you very much to point out the important problem. The main research background of this paper is based on the serious gas leakage problem in coal seam gas extraction, and reasonable sealing material is the key parameter to reduce gas leakage. Therefore, this paper mainly focuses on the influence of the new materials on the gas extraction effect, and compares the effect of the traditional materials on the gas extraction effect.

Comment 5: There are too many chapters in the results and discussion section, resulting in a small amount of content in each chapter. It is suggested to reduce the chapter amounts a bit.

Response: Thank you for your valuable and thoughtful comments. According to the comments, the authors have reduced the chapter amounts a bit.

Comment 6: Although there is “microscopic hydration mechanism” words in the title, the relevant text introduction is very little (only more than 20 lines of concentrated introduction), Should the authors need to consider modifying the title or adding relevant content research?

Response: Thank you for your valuable and thoughtful comments. According to the comments, the authors carefully considered the relationship of the paper, and the authors have changed the title to Analysis of the Micromorphology.

#2 Responds to the Reviewer’s comments:

Comment 1: Some references need to be updated so that authors can keep up with the latest research findings.

Response: Thank you for your valuable and thoughtful comments. According to the comments, the authors have revised the references.

Comment 2: The principle of gas leakage is well explained in Fig.1, however the author's description is not detailed enough. Please refer to the arrow description in the picture.

Response: Thank you for your valuable and thoughtful comments. According to the comments, and combined with the comments given by Reviewer’s 2 comments, the author integrates the chapter with the Introduction section.

Comment 3: In “Gypsum and lime ratio determination” section, for the mechanical properties of the material, why only 3 days mechanical parameters are given in Table 1 and Fig. 3.

Response: Thank you for your valuable and thoughtful comments. Table 1 and Fig 3 are the data obtained to study the optimal ratio of gypsum to lime. The change rule of the previous 3 days can be used to obtain the optimal mixing ratio. Therefore, only 3 days mechanical parameters are given in Table 1 and Fig. 3.

Comment 4: When the author explained the material mechanism, the author mentioned ettringite and other substances. Why didn't the author mark them in Fig. 4, for example, Fig 4 (a1).

Response: Thank you for your valuable and thoughtful comments. According to the comments, the authors have marked ettringite in the corresponding position in Fig. 4.

(b) 3d (×8000) (b1) 28d (×8000)

(c) 3d (×30000) (c1) 28d (×30000)

Fig 3. Scanning electron micrograph of the sealing material at different stages of ageing

Comment 5: In “Analysis of the mechanical properties of sealing materials” section, the authors have done the test of sealing material, but did not analyze the effect of confining pressure on mechanical properties. Please add it.

Response: Thank you for your valuable and thoughtful comments. According to the comments, the revised portion is shown in this paper with red color. The revisions are also shown as follows:

As shown in Fig. 6, with the increase of confining pressure, the compressive strength of the sample increases. The plastic deformation characteristics are more obvious, when the sample reached the peak, no obvious strain softening was observed, the gas sealing material shows great plastic deformation. 

Comment 6: The conclusions should be simplified and summarized, please revise it.

Response: Thank you for your valuable and thoughtful comments. According to the comments, the authors have revised the conclusions, the revised portion is shown in this paper with red color.

Comment 7: Some sentences have spelling mistakes, please correct carefully.

Response: Thank you for your valuable and thoughtful comments. According to the comments, the authors have carefully revised the sentences.

 

Second revision

Responds to the ACADEMIC EDITOR’s comments:

Comment 1: figures 2 and 3: sample age is an unclear terminology. Figures 9 and 10: the number in X-axis is together. They should be separated for each range.

Response: Thank you for your valuable advice. Figure 9 and 10 correspond to Figure 8 and 9 after the revised manuscript. According to the second revision and the final confirmed figures, the authors make the following reply to the ACADEMIC EDITOR.

Fig 2. Compressive strength curves of the sealing material at different curing ages

Fig. 3. Scanning electron micrographs of the sealing material at different times of curing ages. 

Fig 8. Comparison of the gas pressure using conventional polyurethane sealing material and the novel dual-liquid sealing material.

Fig 9 Gas concentration frequency statistics comparison chart

Comment 2: Figure 6(b): Dring depth? Is it 'Drilling depth'?

Response: Thank you for your valuable advice. It is Drilling depth'. The revision is as follows:

(a)

(b)

Fig. 6. Relationship between the drilling depth and the cuttings quantity at the 10 selected sampling sites of the 23302 working face

Comment 3: Both English and presentations should be further improved.

Response: Thank you for your valuable advice. The author further requested that the polishing agency complete the revision of English and presentations. The proof of modification is as follows:

 

Third revision

Responds to the ACADEMIC EDITOR’s comments:

Comment 1: We note that several of your files are duplicated on your submission. Please remove any unnecessary or old files from your revision, and make sure that only those relevant to the current version of the manuscript are included.

Response: Thank you for your valuable advice. The author has deleted the original file and submitted the latest one. However, the old files were deleted, the system sometimes automatically displays the deleted file.

Comment 2: We note that the grant information you provided in the ‘Funding Information’ and ‘Financial Disclosure’ sections do not match.

Response: Thank you for your valuable advice. The author has ensured the match between ‘Funding Information’ and ‘Financial Disclosure’ sections. The updated Financial Disclosure Statement is as follows:

This study was supported by the Science and Technology Project of Henan Province (222102320004), the project of Henan Key Laboratory of Underground Engineering and Disaster Prevention (Henan Polytechnic University) (722403/018/001), the National Natural Science Foundation of China (52174073) and the Outstanding Young Scientist of Beijing (BJJWZYJH01201911413037).

 

Fourth revision

Comment 1: Please amend your Response to Reviewers letter to include a point by point response to each of the points made by the Editor and / or Reviewers. Please follow this link for more information: http://blogs.PLOS.org/everyone/2011/05/10/ how-to-submit-your-revised- manuscript/

Response: Thank you for your valuable advice. The author has revised the comments point by point response to each of the points made by the Editor and / or Reviewers.

---

## [Editor Report · Decision Letter 3]

27 Mar 2023

Experimental characterization and field application of a novel dual-liquid gas leakage material: mechanical properties and microscopic hydration mechanism

PONE-D-22-23213R3

Dear Dr. Hong,

We’re pleased to inform you that your manuscript has been judged scientifically suitable for publication and will be formally accepted for publication once it meets all outstanding technical requirements.

Kind regards,

Jianguo Wang, PhD

Academic Editor

PLOS ONE
---

## [Editor Report · Acceptance letter]

3 Apr 2023

PONE-D-22-23213R3 

Characterization and field application of a novel dual-liquid gas leakage material: Mechanical properties and microscopic hydration mechanism 

Dear Dr. Hong:

I'm pleased to inform you that your manuscript has been deemed suitable for publication in PLOS ONE. Congratulations! Your manuscript is now with our production department. 

Kind regards, 

on behalf of

Dr. Jianguo Wang 

Academic Editor

PLOS ONE